# AFM for Nanomechanical Assessment of Polymer Overcoatings on Nanoparticle-Decorated Biomaterials

**DOI:** 10.3390/nano14181475

**Published:** 2024-09-11

**Authors:** Jonathan Wood, Dennis Palms, Ruvini Dabare, Krasimir Vasilev, Richard Bright

**Affiliations:** 1Future Industries Institute, University of South Australia, Mawson Lakes, Adelaide, SA 5095, Australia; jcwood@swin.edu.au (J.W.); ruvini.dabare@unisa.edu.au (R.D.); 2College of Medicine and Public Health, Flinders University, Bedford Park, SA 5042, Australia; dennis.palms@flinders.edu.au

**Keywords:** atomic force microscopy, AFM, nanoparticles, MePPOx, lateral force microscopy

## Abstract

Nanoparticle adhesion to polymer and similar substrates may be prone to low nano-Newton forces, disrupting the surface bonds and patterning, potentially reducing the functionality of complex surface patterns. Testing this, a functionalised surface reported for biological and medical applications, consisting of a thin plasma-derived oxazoline-based film with 68 nm diameter covalently bound colloidal gold nanoparticles attached within an aqueous solution, underwent nanomechanical analysis. Atomic Force Microscopy nanomechanical analysis was used to quantify the limits of various adaptations to these nanoparticle-featured substrates. Regular and laterally applied forces in the nano-Newton range were shown to de-adhere surface-bound gold nanoparticles. Applying a nanometre-thick overcoating anchored the nanoparticles to the surface and protected the underlying base substrate in a one-step process to improve the overall stability of the functionalised substrate against lower-range forces. The thickness of the oxazoline-based overcoating displayed protection from forces at different rates. Testing overcoating thickness ranging from 5 to 20 nm in 5 nm increments revealed a significant improvement in stability using a 20 nm-thick overcoating. This approach underscores the importance of optimising overcoating thickness to enhance nanoparticle-based surface modifications’ durability and functional integrity.

## 1. Introduction

Nanoparticle-like surfaces can control biological-based interactions by mediating gene expression [1], biodiagnostic assays [2], protein adsorption [3], and enhancing biological imaging [4]. Other applications of nanoparticle-featured surfaces include catalysis and information storage. Titanium oxide nanoparticle surfaces have been used in pigments, sensors, and solar cells. Zirconium oxide nanoparticle surfaces have been used as oxygen sensors in car engines and furnaces. Surface-bound nanoparticles can change the phobic/phallic properties of localised liquids, which can be exploited for numerous functions [5]. Various methods have been reported for the construction of these layered materials [6,7]. A simple and rapid method for producing a highly tunable and defined nanoparticle-enriched surface involves the attachment of carboxylic acid-functionalised gold nanoparticles (AuNPs) to a plasma-derived polymethyloxazoline (MePPOx)-coated substrate. After synthesis via the citrate reduction method, 68 nm-diameter AuNPs were dispersed over a 20 nm-thick MePPOx film immersed in the NP solution, forming amide bonds between the oxazoline structures of the substrate and the carboxylic acid on the AuNPs [8]. Much research has been performed on the chemistry, functionalisation, and techniques of this and similar processes to ensure adequate adhesion of NPs to a plasma polymer-coated substrate [9,10,11,12]. Atomic Force Microscopy (AFM) has been used to manipulate and measure the force required to move individual NPs over various controlled surfaces [13,14,15,16,17]. AFM has characterised and manipulated NPs on varied surface materials to create small regions or ordered NP arrays [18,19,20,21]. While nanomechanical analysis has studied this action on stiff and flat surfaces, minimal analysis has been performed on stabilising these NPs on functionalised or low-modulus surfaces. The movement of large numbers of NPs on biomaterial surfaces is likely to affect their intended application and lower the quality of their performance, not to mention the potential health risks from the removal of individual NPs or clusters of NPs [22,23,24]. To prevent nanonewton (nN) range forces from damaging the integrity of the tailored biomaterial surface, a one-step process was used to ensure the stability of the AuNPs on the MePPOx substrate [25]. Applying a plasma polymer overcoating to the entire surface maintained biological compatibility while creating a chemically homogenous surface. This technique for securing NPs to the substrate with a thin overcoating can be used on other NP-featured surfaces, enhancing overall stability depending on the end application requirements. These include medical applications such as sensors, diagnostics, antibacterial coatings, and drug delivery or reaction systems [26,27].

## 2. Methods and Materials

### 2.1. Sample Preparation

Plasma polymerisation using a 2-methyl-2-oxazoline (Sigma-Aldrich, St. Louis, MO, USA) precursor was achieved via deposition onto a 13 mm-diameter glass coverslip (ProSciTech, Kirwan, QLD, Australia) in a custom-built 13.56 MHz plasma reactor. The glass coverslips were cleaned with acetone and then ethanol and dried under nitrogen gas. Under an 8 × 10^−2^ mbar pressure at a power of 50 W, a 20 nm MePPOx base coating was applied to the precursor material on the glass coverslip. Colloidal gold NPs were synthesised in an aqueous solution. The NPs were boiled and stirred to dissolve 0.01% tetrachloricouric acid (HAuCl_4_, Sigma-Aldrich, St. Louis, MO, USA) and the reducing agent trisodium citrate (Na_3_Ct). A 1% trisodium chloride solution was adjusted to 0.3 mL to synthesise 68 nm AuNPs. The capping agents mercaptosuccinic acid (MSA, Sigma-Aldrich, St. Louis, MO, USA) and sodium hydroxide (Sigma-Aldrich, St. Louis, MO, USA) stabilised the NPs. Immersion of the MePPOx-coated coverslips in colloidal AuNP solution for up to 24 h immobilised the NPs onto the MePPOx surface. MSA produced a stable Au–S bond with the AuNPs. After citrate reduction, the AuNPs formed a carboxylic acid monolayer, promoting electrostatic repulsion between the AuNPs and carboxylic groups bound to the Au. MePPOx–AuNP binding occurred by forming amide bonds via a reaction between the oxazoline and monomer. This reaction formed a cationic centre and ring-opened oxazoline on the MePPOx [8,28]. An additional MePPOx coating was applied in a further process using the plasma reactor. The overcoating thickness was a time-dependent parameter occurring at one minute for a 10 nm overcoat, two minutes for a 15 nm overcoat, and three minutes for a 20 nm overcoat. A range of plasma-derived base coatings and different-sized AuNPs were trialled; these are contained in the Appendix A. The MePPOx base film and 68 nm AuNP variation were chosen due to the higher measured adhesion and friction forces between the NPs and the base surface, which indicates stronger bonding.

### 2.2. Atomic Force Microscopy (AFM)

AFM analysis was performed in the air using a JPK NanoWizard III (Bruker, Billerica, MA, USA) connected to a Nikon Eclipse Ti series inverted optical microscope (Nikon, Shinagawa-ku, Tokyo, Japan). Linux-based JPK control v5 software was used to access instrument controls and collect data. Data analysis was performed using Gwyddion scanning probe microscopy version 2.65 (http://gwyddion.net/download.php, accessed on 29 January 2024), ImageJ version 1.54i (NIH, Bethesda, MD, USA), and Office Excel 365 ProPlus version 2311 Build 16.0.17029.20140 (Microsoft, Redmond, WA, USA). NT-MDT NSG03 silicon nitride cantilevers (NT-MDT Spectrum Instruments, Zelenograd, Moscow, Russia) were used, with a measured spring constant between 2 and 3 N/m and a guaranteed tip radius of less than 10 nm. Cantilever calibration was achieved through a force curve procedure on a glass microscope slide. Gwyddion software was used to determine the deflection slope of the curve to determine the cantilever sensitivity, Q-factor, Frequency, amplitude, and spring constant. Thermal tuning of the cantilever was explicitly performed for amplitude modulation (AM) mode operation.

Once set, the tip contacted the sample surface for topographical scanning [29]. AM and contact mode topography were performed using AM mode topography to oscillate the cantilever via a z-axis piezo over a preset amplitude. This mode intermittently contacted the cantilever tip to the sample surface as it rastered over a preset scan area. Contact mode topography moved over a preset surface area with the cantilever tip in constant contact with the sample surface. A set z-axis force was applied by the tip on the surface substrate, which was the normal force and was expected to be in the substrate plane. Contact mode applied both z-axis and lateral or x-axis forces on the surface as the cantilever scanned [30]. In contact mode, the cantilever tip experienced short-range repulsive forces, while in AM mode, it encountered both attractive and repulsive forces, which influenced the amplitude and Frequency of cantilever oscillation, influencing the phase. The phase signal measures the energy dissipated between the tip and sample surface, determining the viscoelastic, chemical, and adhesive properties. Energy dissipation contrast could separate material compositions across the scan area. Phase contrast showed the chemical distinction between the AuNPs and the MePPOx film [31,32,33].

### 2.3. LFM Mode

Lateral Force Microscopy (LFM) measures surface friction alongside contact mode topography through lateral deflection trace and retrace data. Data from this mode require postprocessing in analysis software such as Gwyddion and Microsoft Excel. The torsional deflection of the cantilever evaluates friction as it scans in contact mode. Increased friction is related to increased cantilever torsion from chemical or topographical changes across the substrate. By comparing opposing scan directions, a friction loop can be plotted. The friction coefficient can be equated by taking the median between the upper and lower loop values. Friction force values are determined by multiplying the friction coefficient values by the Set Point force [34,35]. Force mapping is a mode in which a preset number of force curves are evenly spaced over a set area of the sample surface. In this mode, the tip moves from a set distance above the surface into surface contact with it at a preset force and velocity value, followed by retraction to its initial position, allowing several nanomechanical values to be measured. Force curve data of the tip–substrate interaction can be derived from this process by fitting individual curves or batch processing over the entire scan area [36]. Values for elastic modulus, adhesion, and stiffness across regions of contrasting composition and structures can be derived, which further characterise surface functionalization and interactions.

### 2.4. Statistical Analysis

Data analysis and visualisation were conducted using Gwyddion software version 2.65 [37], Microsoft^®^ Excel^®^ for Microsoft 365 MSO (Version 2311 Build 16.0.17029.20140), and GraphPad Prism version 10.1.0 [38]. All experiments were performed in triplicate unless otherwise stated. Results are presented as the mean and standard deviation (SD), and a *p*-value of less than 0.05 was deemed statistically significant.

## 3. Results and Discussion

### 3.1. NP Resolution of Overcoating

AFM topography profiles of the AuNPs imaged using a conical-shaped cantilever tip showed a lateral increase in the overall width of the NPs, which is related primarily to the side angle, often termed the tip convolution effect [30,39]. The NT-MDT NSG cantilever has a reported side angle of 18° and a guaranteed tip radius of less than 10 nm, as quoted by the manufacturer, which needs to be considered in the resolution of any NP, especially for features of comparable size to the cantilever tip. A lateral increase is apparent in the topographical images of near spherical NPs, where the initial tip sidewall contacts the tip at the furthest point from the centre of the NP [40], as shown in Figure 1. An added plasma coating is unlikely to attach cleanly to the region between the underside of the spherical NPs and the substrate directly below. The thicker the applied coating is, the wider the spread and ‘averaging’ of defined and complex surface features [41].

As verified in Figure 2, the application of overcoating on these NPs is expected to increase the width as the coating thickness increases. Therefore, the thicker the coating, the less defined the structure becomes [42]. Figure 2A,C visually shows this width increase, with (A) being an AC mode scan of the 68 nm diameter NP with a 10 nm MePPOx overcoating, (B) with a 15 nm overcoating, and (C) with a 20 nm overcoating. An applied overcoating via plasma deposition covers both the base substrate and the NP. Therefore, the expected increase in height would be close to zero, which is not the case, as shown by the NP height vs. width plot in Figure 2D, where a slight increase in average NP height occurs with an increase in overcoating thickness. An average 5 nm increase in NP height is observed with a doubling overcoating thickness from 10 to 20 nm. A combination of factors can lead to a slight increase in height, such as minor variances in overcoating adhesion between the existing MePPOx substrate and the peak of the NP due to surface geometry, bonding chemistry, and short-range forces [43,44].

The lateral increase in the NP morphology is significant, with an approximately 30 nm increase in width resulting from a doubling of the overcoating thickness. Figure 2D shows the relationship between the increase in width to height and the thickness of the applied coating. A 30 nm increase in width compared to a 5 nm increase in height, when transitioning from 10 to 20 nm overcoating, results in an expansion ratio of 6:1. This factor needs to be accounted for if NP size or patterning morphology is a critical factor for the functionality of the surface. A uniform overcoating covering the entire AuNP requires a moderate level of spreading and ‘smoothing’ of the NP’s topography as the overcoating thickness approaches a reasonable ratio to the NP height. A thicker overcoating is also expected to be less flexible in detailing acceptable nanofeature resolution and will likely fail to adhere closely to the base of the sphere because the elastic resistance of the polymer will be stronger than the reduced adhesion area [41]. Negligible differences were noted in the root-mean-square (RMS) and average roughness (Ra) values among scans with different overcoat thicknesses. These values were acquired in AM mode. Overall, 3.3 nm and 4.1 nm nonlinear variations in the RMS and Ra roughness were measured, as shown in Table 1. This variation is partly related to differences in the number of particles in the scan region. Any changes in roughness between the AuNPs exposed to the MePPOx coating were minimal and related to the z-axis height of the NPs relative to the base substrate. Despite the increase in the width of individual NPs through the application of thicker coatings, changes in the z-axis height were minor.

Elastic modulus measurements were acquired through force curve mapping over two orders of magnitude of Set Point force applied by the cantilever tip on the centre of an AuNP across three samples. Sample one possessed no MePPOx coating, sample two featured a 10 nm overcoating, and sample three had a 20 nm overcoating. As shown in Figure 3A, minor and variable differences in elastic modulus values were apparent for applied forces in the low nN range up to 20 nN, beyond which a more apparent division appeared. A higher modulus for the noncoated NP was due to the reduced volume of the softer, deformable MePPOx polymer. Overcoating the NPs with 10 and 20 nm particles reduced the overall modulus by increasing the volume of the lower modulus, deformable MePPOx material, as shown in the representations in Figure 3B–D.

### 3.2. NP Adhesion

The contact mode topography applies a lateral force on the NPs as the tip rasters over the surface. The magnitude of this force is dependent on a combination of the cantilever spring constant and the Set Point, which applies a z-axis force between the cantilever tip and the sample surface [17,45]. During contact mode scanning, the cantilever tip drags across the MePPOx substrate until the tip contacts the side of the NP. Any raised surface feature can be viewed as a physical barrier, interrupting the AFM hardware’s attempts to maintain a continuous scan velocity. As the tip drags up the side of the NP, the cantilever is still forced to continue in the x-axis scan direction, which applies an increasing lateral force on the NP by the tip. When the lateral force is high enough, the bonds between the MePPOx substrate and the AuNPs can be broken, and the NPs move across the surface, as shown in Figure 4A. The application of a MePPOx overcoating creates additional anchoring of the NPs to the substrate, which requires greater lateral forces to break, as shown in Figure 4B. In addition, the overcoating can reduce the lateral force applied at any point on the side of the NP, lowering the overall force by decreasing the feature’s side angle, which in turn lowers the lateral force at any one point by decreasing the angle of the x-axis force and increasing the z-axis force applied by the tip. Extending the length of the sidewall with a thicker coating also reduces the lateral force from the tip by spreading the force over a more extended area. Figure 4C shows a 2D plot of an NP with no applied coating. The side angle of the NPs, as measured using ImageJ software, was 79°, while the 20 nm overcoated NPs, as shown in Figure 4D, possess a side angle of 64°.

The measurement of the force required to break the bonds that hold the NPs to the MePPOx surface was not straightforward, as a set lateral force was observed to be able to move some NPs but not others. The ratio of NPs that moved was different from those that did not depend on controllable parameters such as the Set Point force and the coating thickness, and difficult-to-control parameters such as the number of surface–NP bonds. Averaged at 68 nm ± 5 nm in diameter, the size variance and related contact area of the individual NPs with the base substrate are not expected to vary significantly. Localised roughness and topographical anomalies of the MePPOx substrate and the individual AuNPs may vary enough to dictate the number of surface–NP bonds [17]. Even at typical force values >20 nN, a small percentage of NPs adhered to the substrate. A small percentage of NPs (<10%) appeared to move at meagre forces, even with a thin overcoating of ≤10 nm. These are most likely residue NPs that were not removed by postprocessing rinsing in flowing Milli-Q water. Any NP movement over the surface area by applying a small external force is potentially damaging to the nanopatterning order of the surface. Patterning changes were visible when the number of moved NPs exceeded 15–20%. Apparent changes are observed in the significant gaps from displaced NPs and clumping regions caused by the tip pushing many NPs into clusters, often occurring at the scan edges and creating mass surface features several hundreds of nanometres in diameter. Figure 5A,B shows an AC mode scan and plot across a region of clumped NPs greater than 300 nm in length after a small contact mode scan. The high volume of NPs in these clumped regions can be contrasted with the large region containing almost no NPs. Figure 5C,D plots five craters left in the MePPOx substrate after the movement of individual NPs by the cantilever tip, with crater widths, each measuring approximately 120 nm in diameter. The scan in Figure 5C not only displays the absence of AuNPs after a contact mode scan of a surface with no overcoating but also shows visible damage to the MePPOx substrate, present as craters where the NPs adhered to the substrate. These craters measure a depth of approximately 5 nm.

Contact mode measurements were performed on five individual craters. These craters are left in the base MePPOx material after an AuNP was removed and shifted to the edges of the scan area. The five craters’ average value was calculated and is shown in a bar graph, Figure 6A. Each bar in Figure 6A represents a sample with a particular overcoating thickness, ranging from no overcoating to 20 nm MePPOx overcoating. The general trend displayed in the bar chart is that the thicker the overcoating, the reduced the depth of the crater (*p* < 0.05). The cause of this is supported by Figure 6B, where the thicker overcoating exhibited a trend to reduce the crater area. The movement of an AuNP with an overcoating causes the nanoparticle to break free before being moved across the scan area by the cantilever tip. The overcoating material is expected to settle and partially close over the crater, as supported by the images in Figures 8–12, displaying craters and no raised MePPOx openings. With a thicker overcoating, such as the 10–20 nm MePPOx, the material better ‘fills in’ the gap.

### 3.3. NP Stability with an Overcoating

Alternative methods were explored to increase the NP–substrate adhesion strength while lowering the thickness of the MePPOx coating. Sintering the MePPOx base film before overcoating was expected to increase the contact area between the NPs and the base substrate through fusion [46,47,48]. Heating gradients need to be controlled so as not to substantially change the surface chemistry or surface topography, which is vital in the case of polymers such as MePPOx, where the glass transition temperature is approximately 80 °C [49]. Excessive temperatures may result in distortion or evaporation of the substrate material, leading to a heterogeneous surface, thus causing the opposite of the intended effect by lowering the contact area between the NP and MePPOx substrate. Initial trials involving the sintering of a 20 nm MePPOx film on a glass coverslip patterned with AuNPs at a density of approximately 25 particles/µm² resulted in reduced NP adhesion to the base substrate. This experiment involved sintering non-overcoated AuNP samples in an SEM tube furnace at temperatures between 200 °C and 400 °C for 30 to 60 min. This temperature was above the glass transition temperature and the 110 °C annealing temperature of MePPOx [50]. An AM contact mode topography process verified reduced adhesion, which was performed throughout this paper to test the surface stability of the AuNPs. The results revealed a significant disruption in the temperature and heating time of the MePPOx–NP substrate, with an applied Set Point force of 15–20 nN. Figure 7A–J displays the results for a non-overcoated sample sintered between 200 °C and 400 °C for 30 min. NP movement was easily observable and appeared to increase at higher temperatures and heating times up to 400 °C. Lower sintering temperatures (<200 °C), significantly below the annealing and glass transition temperatures, may increase NP bonding; however, the consistency of stable and uniform heating at such a low value is likely to be challenging and prone to significant errors in repeatability [51,52].

### 3.4. NP Adhesion to Overcoating Thickness

NP adhesion testing was performed over 5 nm overcoating thickness intervals with graduating Set Point forces in contact mode to determine NP stability. Due to the difficulty in determining the absolute value of the Set Point force required to move NPs, not all particles moved at the same value. Standardisation needs to be set at a point where functionality concerning NP patterning is likely to be impeded. A value to set the minimum number of NPs per unit area that can be moved before surface functionalisation is compromised appeared as an evident visual change when NP displacement exceeded 10% of the number of particles in the scan region. Once the number of displaced NPs reached 15–20%, more extensive regions of NP clumping appeared, and larger NP-vacant areas were present. Once regions of clumping exceeded 10 NPs, surface order and density consistency were compromised, with NP stacking occurring in association with NP-vacant surface regions exceeding 500 × 500 nm. Figure 8A–F displays an array of tapping mode scans acquired after a smaller area contact mode scan was performed at a particular Set Point—a sample with no applied coating provided control of the effect of the cantilever tip on the NPs. Starting at a Set Point of 1 nN, Figure 8A, small areas of NP clumping and separation appeared even at this level of the applied force from the tip dragging across the surface. Minor clumping regions were present due to sample preparation. As the Set Point increases to 100 nN, Figure 8F, larger areas of clumping in the region of the previously performed contact mode scan became visible, as indicated by the white arrows.

Increased clumping is evident on the 5 nm overcoating samples compared to those with no overcoating, as shown in Figure 9A–F. This clumping is most likely due to the very thin overcoating being damaged around the edges of the contact mode scan, where the cantilever tip stops and turns around. Damage to the thin MePPOx film is supported by the phase scan images of the base MePPOx shown in Figure 14. This damage is likely to either act as a catalyst in moving NPs as the thin overcoating moves, or the damage becomes a barrier that stops and traps NPs on the uneven surface of the overcoating.

From 5 to 10 nm, a slightly thicker overcoating appears to display less NP clumping, as shown in Figure 10A–F. Comparing NP movement in Figure 9 to Figure 10, reduced NP movement and clumping are expected owing to thicker overcoating, which reduces damage caused by the cantilever tip. At this thickness, damage appears less likely to occur, lowering NP clumping at the scan edges. Clumping at this overcoating thickness is noticeable in Figure 10C, where the cantilever force is 10 nN. The gaps between the NPs begin to appear in Figure 10C and continue to be evident in Figure 10F as higher tip forces are applied.

Figure 11A–F displays samples with a 15 nm MePPOx overcoating scanned at Set Points ranging from 1 nN to 100 nN. Clumping is again noticeable in Figure 11C at the same applied tip force. The marginal difference in NP layer protection between the 10 and 15 nm overcoatings is apparent.

Post-contact tapping mode scans of a sample with a 20 nm MePPOx overcoating are displayed in Figure 12. Post-contact tapping mode scans of a sample with a 20 nm applied MePPOx overcoating are shown at a reduced scan area of 1.5 × 1.5 µm, rather than the 5 × 5 µm scan area shown in Figure 8, Figure 9, Figure 10 and Figure 11. The scan area was reduced to highlight the increased NP stability over the contact scan area. Images in Figure 12 demonstrate the influence of the cantilever tip on the 68 nm AuNPs from a smaller area contact mode scan previously performed in the centre of the image at a Set Point from Figure 12A of 1 nN to Figure 12F at 100 nN. There was minimal movement or clumping across the Set Points ranging from 1 nN to 100 nN. The 20 nm coating appears to be the optimal surface treatment for preventing the movement of the AuNPs.

Figure 13 displays a series of 5 × 5 μm tapping mode height scans, following a 2 × 2 μm area previously scanned using contact mode. The x-axis image progression in Figure 10 displays an increasing Set Point force applied by the cantilever tip on the surface in contact mode, ranging from 1.0 to 100 nN. The y-axis images show 5 nm changes in an applied MePPOx overcoating thickness. Samples with a 5 nm and no applied overcoating show increased NP vacancies and clumping, even at very low nN forces. For 10 and 15 nm applied overcoatings, a 15–20% NP movement and 10+ particle clumping regions occur at forces around 10 nN. This level of overcoating thickness does appear to reduce NP movement under low nN laterally applied forces. A significant increase in NP stability is apparent when using a 20 nm overcoat. NP stability at this overcoating thickness is maintained even when subjected to an applied force of over 50 nN by the cantilever tip. The difference between the application of a 15 nm and a 20 nm overcoating displays a rise in NP stability under an applied force of more than five times greater. A 20 nm overcoating ensured a notable rise in securing the surface consistency, even under low nN-range forces.

Phase scans generally display higher resolution than topographical height scans because the phase interlinks with changes in cantilever amplitude. In this case, the phase is related to dampening and adhesion due to damage to the lower-modulus MePPOx film. The change in energy dissipation caused by the tip penetration depth into the 20 nm thickness of the base MePPOx film contrasts with the stiffer glass substrate underneath. Deeper tip penetration into the substrate creates a more significant lag in tip oscillation, which is observable in the phase contrast. Due to this effect, a lower measured elastic modulus is related to a more remarkable phase shift. Therefore, as the MePPOx is damaged, regions with considerable thickness variations in such a thin film change the elastic modulus [33]. Figure 14A–E clearly shows the progressive damage to the MePPOx surface of a non-overcoated AuNP sample. Damage to the substrate strongly contributes to the deadhesion of the NPs. When tip–sample forces reach 20 nN and above, the damage becomes significant to the point of tip ploughing through the MePPOx. Figure 14F–J shows a sample with a 20 nm coating. Negligible damage to the MePPOx surface or movement of the NPs occurs on the sample, with tip Set Point forces ranging from 1–100 nN. A noticeable difference in tip-induced damage is displayed when comparing the application of the lateral forces during contact mode imaging of the 20 nm overcoated and non-overcoated samples. A clear advantage is apparent in applying an overcoating of a certain thickness to a surface functionalised with NPs, particulates, or similar nanostructures. Applying a 20 nm coating of stabilised and fixed NPs to the substrate protects the base substrate from significant damage during surface analysis and end application.

The AFM is generally limited to testing under low applied forces, up to the low micro-Newton scale. Operation of the surface under higher forces or contacts would require pin-on-disk or Coefficient of Friction instrumentation. Advanced parameters like Sbi, Sci, Svi, and Sdr can offer valuable insights in specific contexts; however, they were outside the scope of this study. Additionally, further testing for biomedical applications requires evaluating the material’s durability and reactions within biological fluids. Specifically, testing in a liquid environment, such as a saline-based solution, is essential to assess electrochemical reactions and changes to surface bonding. This testing is crucial for ensuring the material’s functionality and stability in its intended biological setting.

## 4. Conclusions

NP-patterned surfaces designed for biological applications must be exposed to interactions within harsh biological environments. Due to this, the level of adhesion and stability of the NPs to the base substrate must be quantified. Hopefully, this will prevent disassociated NPs from being freely released into the surrounding environment and minimise reduced surface functionality caused by significant changes to NP patterning. A biologically compliant overcoating is a straightforward, one-step solution to increase NP adhesion while minimally altering the surface topography. A MePPOx film was chosen as the base substrate due to its biocompatibility and strong bonding chemistry with AuNPs. Application of the same polymer as the base substrate film and as an overcoating layer provided a consistent biocompatible chemistry for the overall surface. Testing the adhesion strength of the NPs was performed using AFM contact mode topography, which allowed a range of forces to be constantly applied to the surface. The AFM is an adaptable instrument that allows force testing of the surface and topographical analysis of the potential changes to the surface. Before and after measurements of the surfaces could be performed, comparing any alterations. The movement of the NPs was found to vary with the range of applied forces.

However, disrupting over 10% of the NPs or forming clumps of more than 10 particles reduced the surface’s pattern-driven functionality. Testing a range of MePPOx overcoating thicknesses determined that a 20 nm-thick MePPOx overcoating is the minimum thickness required to minimise NP disruption and MePPOx damage from interaction forces above 50–100 nN. A 20 nm-thick overcoating increases the NP surface area by approximately 30%, which needs to be accounted for in the surface design. Despite this topographical change, applying an overcoating of at least 20 nm ensures the surface nanopatterning remains functionally stable when applying a greater force than a thinner overcoating. These findings highlight the critical role of precise overcoating thickness in maintaining the integrity and functionality of NP-patterned surfaces in challenging biological environments, paving the way for more robust and reliable biomedical applications. The testing and overcoating solution for binding particles to a surface holds significant potential for use with various biomaterials, particularly those needing a chemically uniform nanostructured surface.

## Figures and Tables

**Figure 1 nanomaterials-14-01475-f001:**
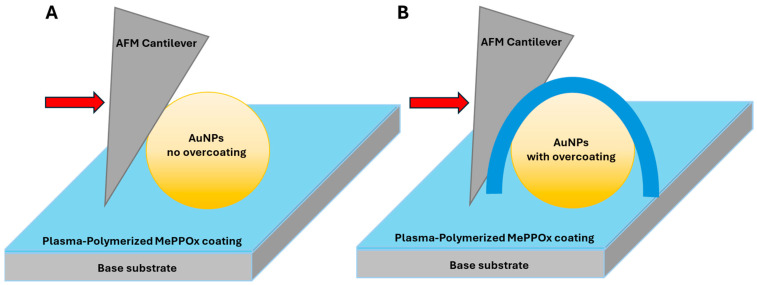
Cantilever tip interaction with a spherical AuNP: (**A**) without overcoating and (**B**) with thick MePPOx overcoating. Red arrows depict the direction of the AFM cantilever.

**Figure 2 nanomaterials-14-01475-f002:**
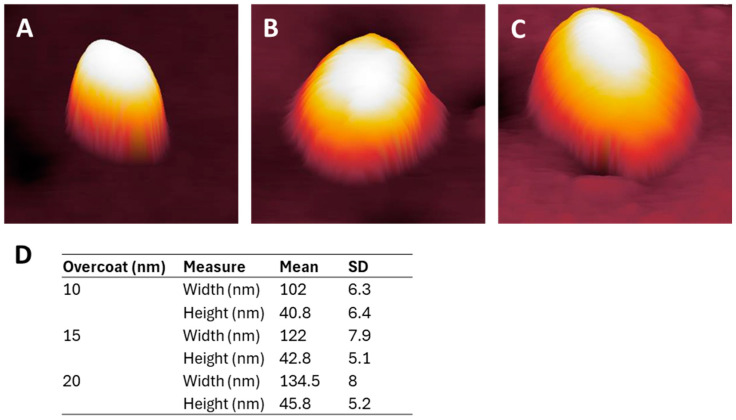
Gwyddion software 3D representations of individual 68 nm AuNPs with MePPOx coatings of (**A**) 10 nm, (**B**) 15 nm, and (**C**) 20 nm. (**D**) Table showing the heights and widths of 11 individual NPs with 10, 15, and 20 nm MePPOx overcoats, presented as mean ± SD, *n* = 5. Cantilever settings are provided in the Appendix A.

**Figure 3 nanomaterials-14-01475-f003:**
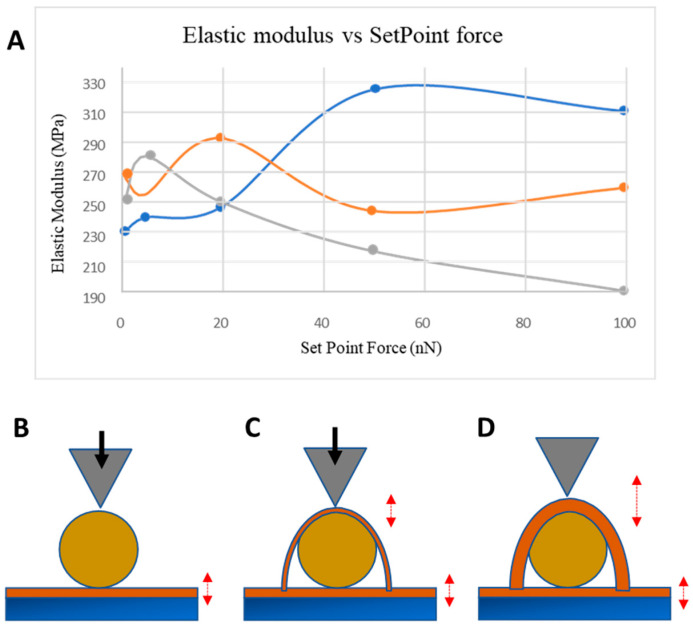
(**A**) 2D plot of the elastic modulus vs. the applied normal force (Set Point) on a sample with no MePPOx overcoating (blue), a 10 nm overcoating (orange), and a 20 nm overcoating (grey). Surface elasticity (shown by red arrows) on gold nanoparticles with a 20 nm MePPOx film layered on a glass base (**B**) without MePPOx overcoating (blue), (**C**) with a 10 nm overcoating (orange), and (**D**) with a 20 nm overcoating (grey). Values for each cantilever calibration and setup are displayed in the Appendix A.

**Figure 4 nanomaterials-14-01475-f004:**
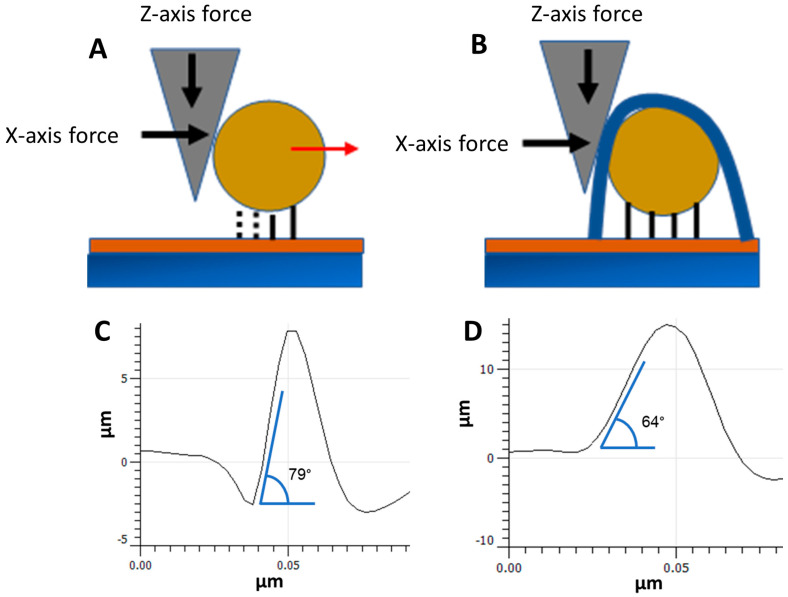
Schematic of the lateral forces applied by the cantilever tip to break the substrate-NP bonds with (**A**) no overcoating, (**B**) applied MePPOx overcoating (blue), (**C**) a 2D plot of AuNPs with no overcoating, and (**D**) a 2D plot of AuNPs with a 20 nm overcoating. Data were acquired using an NT-MDT NSG03 cantilever. Each overcoating measurement was performed using two different cantilevers from the same batch supplied by the manufacturers. Free-air amplitude values at a magnitude of 10^−15^ are below the expected noise floor of the instrument. However, the values presented are from instrumentation software. The cantilever values for (**A**,**C**) are sensitivity 29 nm/V, Q-factor 158, free-air amplitude 7.8 × 10^−15^ m/√Hz, spring constant 2.8 N/m, and Set Point 27 nm. The cantilever values for (**B**,**D**) are sensitivity 25.4 nm/V, Q-factor 157.6, free-air amplitude 8.8 × 10^−15^ m/√Hz, spring constant 2.3 N/m, and Set Point 22 nm.

**Figure 5 nanomaterials-14-01475-f005:**
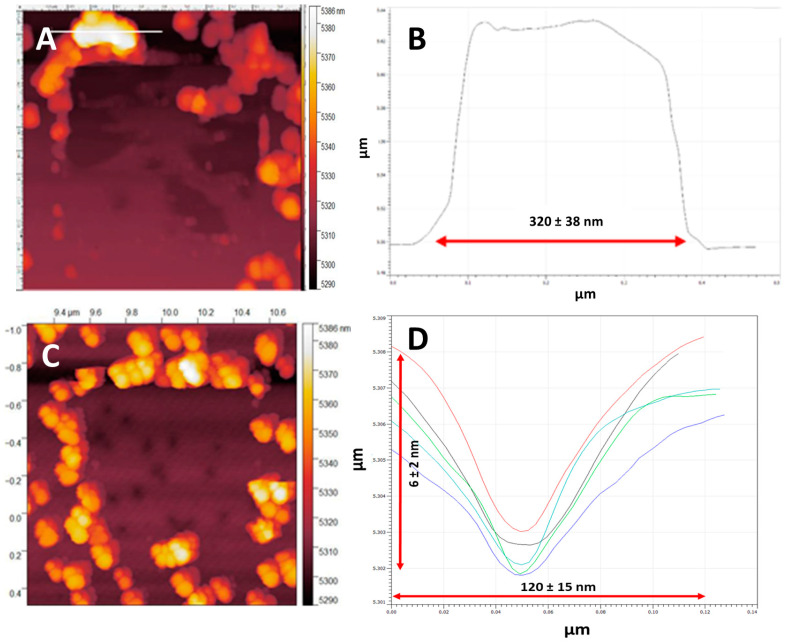
(**A**) NP clumping around the edges of a 5 × 5 µm contact mode scan, (**B**) a 2D plot displaying the diameter of an NP clumping region, (**C**) an AM mode 10 × 10 µm topography scan performed after the contact mode scan, and (**D**) 2D plots over craters left by moved NPs. Each coloured line represents a different crater. Height and width measures are presented as mean ± SD and *n* = 5. Scans were performed using an NT-MDT CSG10 cantilever with the following settings: sensitivity 47 nm/V, Q-factor 35, free-air amplitude 1.7 × 10^−13^ m/√Hz, spring constant 0.14 N/m, and Set Point 38.6 nm.

**Figure 6 nanomaterials-14-01475-f006:**
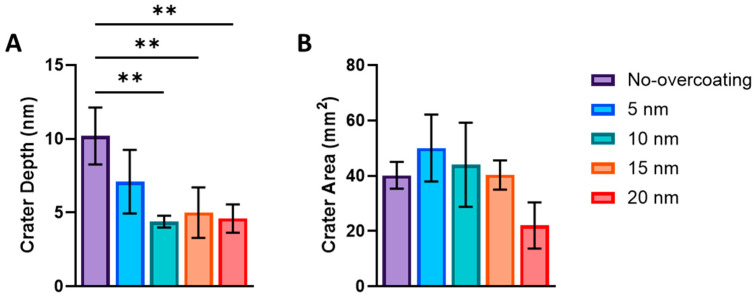
Craters on the MePPOx substrate left after NP movement. Each group was subjected to MePPOx overcoating at 0, 5, 10, 15, and 20 nm thicknesses. Overcoating groupings are compared to (**A**) the depth of the craters (** *p* < 0.01) and (**B**) the surface area of the craters. Data are plotted as mean ± SD and *n* = 3. No significant difference was noted between the plots. Values for each cantilever calibration and setup are displayed in the Appendix A.

**Figure 7 nanomaterials-14-01475-f007:**
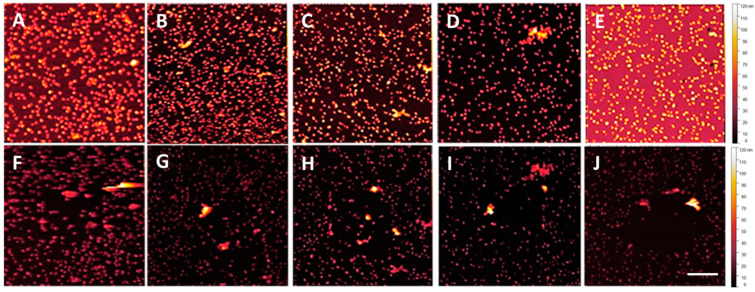
AM mode topography of the MePPOx substrate before (**A**–**E**) and after (**F**–**J**) sintering. The results for sintering at (**F**) 200 °C for 30 min, (**G**) 200 °C for 60 min, (**H**) 300 °C for 30 min, (**I**) 300 °C for 60 min, and (**J**) 400 °C for 30 min. The scale bar in the bottom right of the image represents 1 µm and serves as the scale for all images. Scans were performed using an NT-MDT NSG03 cantilever with sensitivity 27 nm/V, Q-factor 150, free-air amplitude 1 × 10^−14^ m/√Hz, spring constant 1.9 N/m, and Set Point 22.6 nm.

**Figure 8 nanomaterials-14-01475-f008:**
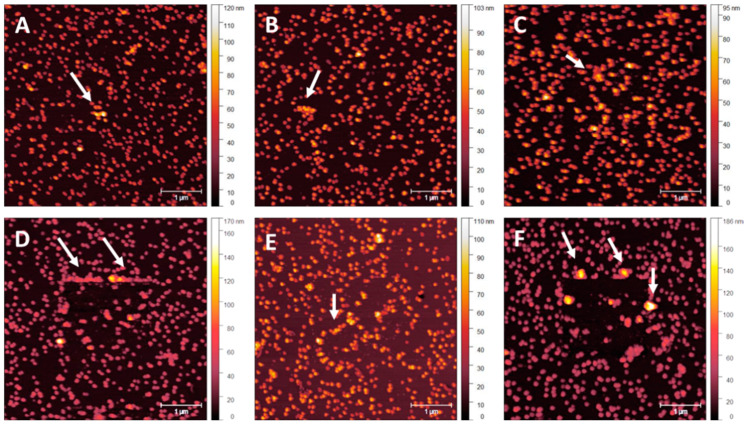
Post-contact tapping mode scans of a sample with no applied MePPOx overcoating. Images (**A**–**F**) show the cantilever tip influence on the 68 nm AuNPs after a smaller area contact mode scan previously performed in the centre of the image at a Set Point from (**A**) 1 nN, (**B**) 5 nN, (**C**) 10 nN, (**D**) 20 nN, (**E**) 50 nN, and (**F**) 100 nN. White arrows highlight NP clumping caused by the contact mode scan.

**Figure 9 nanomaterials-14-01475-f009:**
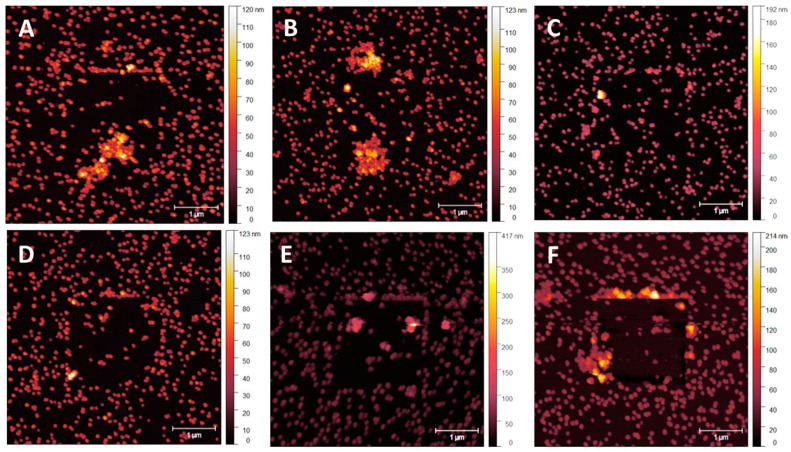
Post-contact tapping mode scans of a sample with a 5 nm MePPOx overcoating. Images (**A**–**F**) show the cantilever tip’s influence on the 68 nm AuNPs from a smaller area contact mode scan previously performed at the centre of the image at a Set Point from (**A**) 1 nN, (**B**) 5 nN, (**C**) 10 nN, (**D**) 20 nN, (**E**) 50 nN, and (**F**) 100 nN.

**Figure 10 nanomaterials-14-01475-f010:**
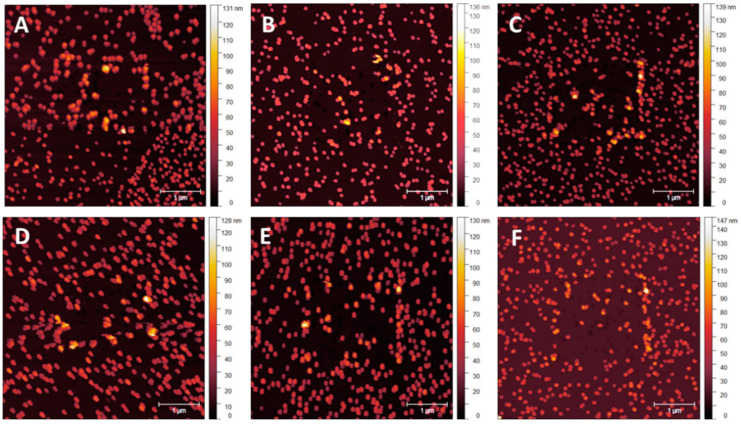
Post-contact tapping mode scans of a sample with a 10 nm MePPOx overcoating. Images (**A**–**F**) show the influence of the cantilever tip on the 68 nm AuNPs from a smaller area contact mode scan previously performed in the centre of the image at a Set Point from (**A**) 1 nN, (**B**) 5 nN, (**C**) 10 nN, (**D**) 20 nN, (**E**) 50 nN, and (**F**) 100 nN.

**Figure 11 nanomaterials-14-01475-f011:**
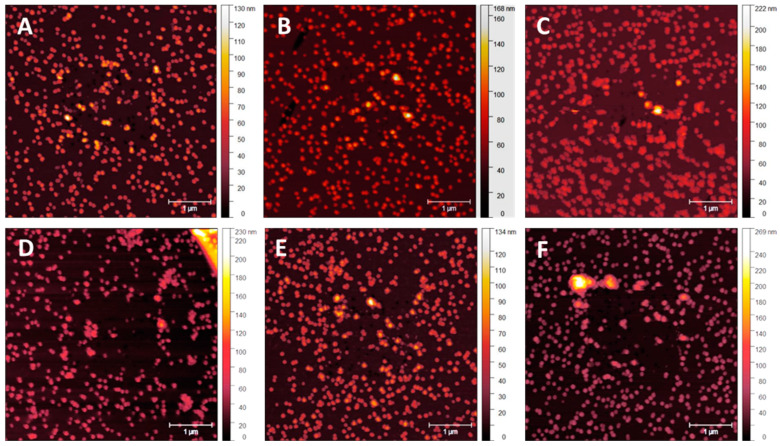
Post-contact tapping mode scans of a sample with a 15 nm MePPOx overcoating. Images (**A**–**F**) show the cantilever tip’s influence on the 68 nm AuNPs in a smaller area contact mode scan previously performed in the centre of the image at a Set Point from (**A**) 1 nN, (**B**) 5 nN, (**C**) 10 nN, (**D**) 20 nN, (**E**) 50 nN, and (**F**) 100 nN.

**Figure 12 nanomaterials-14-01475-f012:**
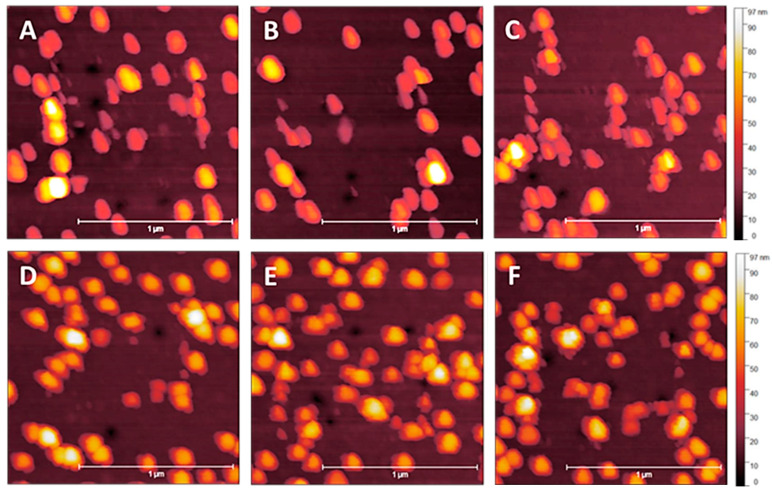
Post-contact tapping mode scans of a sample with a 20 nm MePPOx overcoating. Images (**A**–**F**) show the cantilever tip’s influence on the 68 nm AuNPs from a smaller area contact mode scan previously performed in the centre of the image at a Set Point from (**A**) 1 nN, (**B**) 5 nN, (**C**) 10 nN, (**D**) 20 nN, (**E**) 50 nN, and (**F**) 100 nN. Values for each cantilever calibration and setup are displayed in the Appendix A.

**Figure 13 nanomaterials-14-01475-f013:**
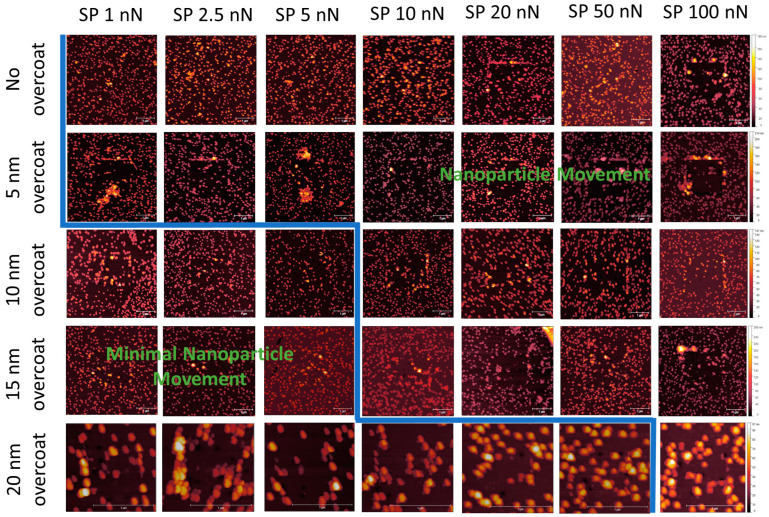
Post-contact AM mode scans comparing NP movement on MePPOx surface film substrates patterned with 68 nm AuNPs with different overcoating thicknesses and applied forces by an AFM cantilever tip.

**Figure 14 nanomaterials-14-01475-f014:**
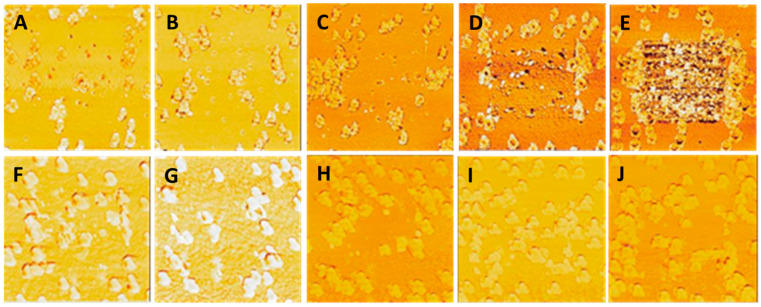
AM mode phase images from post-contact mode scanning at 1.5 × 1.5 µm at Set Point forces of (**A**,**F**) 1 nN, (**B**,**G**) 5 nN, (**C**,**H**) 20 nN, (**D**,**I**) 50 nN, and (**E**,**J**) 100 nN. Images (**A**–**E**) are samples with no applied overcoating, and (**F**–**J**) are samples with a 20 nm overcoating. Scans were performed using an NT-MDT NSG03 cantilever with the following specifications: sensitivity 32.3 nm/V, Q-factor 146, free-air amplitude 1 × 10^−14^ m/√Hz, spring constant 1.6 N/m, and Set Point 25.5 nm. Values for each cantilever calibration and setup are displayed in the Appendix A.

**Table 1 nanomaterials-14-01475-t001:** RMS and Ra roughness measurements over a 5 × 5 µm AM mode scan area from a non-overcoated sample to a 20 nm overcoated sample. Mean ± SD, *n* = 5. Values for each cantilever calibration and setup are displayed in the Appendix A.

Roughness	No Overcoat	5 nm	10 nm	15 nm	20 nm
**RMS (nm)**	14.4 ± 6.6	14.8 ± 3.7	11.5 ± 2.9	13.0 ± 5.9	12.9 ± 7.0
**Ra (nm)**	10.9 ± 4.5	11.1 ± 8.4	7.0 ± 6.5	9.1 ± 3.3	8.5 ± 4.6

## Data Availability

The raw data supporting the conclusions of this article will be made available by the authors on request.

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
