# Peer review of "AFM for Nanomechanical Assessment of Polymer Overcoatings on Nanoparticle-Decorated Biomaterials"

_nanomaterials, 2024, doi:10.3390/nano14181475_

Round 1
Reviewer 1 Report
Comments and Suggestions for Authors
The manuscript is carefully and meticulously written, with the authors providing detailed descriptions of the obtained data. While the overall quality of the article is high, it lacks a robust discussion and a broader approach to the topic. Although many measurements were conducted, the manuscript does not clearly articulate the motivation behind the studies beyond the general notion that maintaining proper topography is crucial for medical applications. The extensive detail provided is both a strength and a weakness. On one hand, it demonstrates thoroughness; on the other hand, it makes it difficult to follow the discussion and grasp the key results, as the significance of the indicated values and the conclusions or hypotheses they suggest are not clearly communicated. More than 10 pages of results (mostly results, as any discussion that exists is not clearly distinguished) are summarized in a single paragraph in the conclusions section, which seems inadequate. In fact, the introduction should more strongly emphasize the aim of the work and the novelty it presents.
From a technical standpoint, there is a minor error on line 180: it references "Figure 2 D and E," but there is no "E" in Figure 2.
Overall, the manuscript is of high quality, and I recommend accepting the article after the suggested corrections are made.
Reviewer 2 Report
Comments and Suggestions for Authors
The manuscript number nanomaterials-3169711, entitled “AFM for Nanomechanical Assessment of Polymer Overcoatings on Nanoparticle Decorated Biomaterials” presents one-step process to ensure the stability of the AuNPs to the MePPOx substrate used to prevent the nanonewton (nN)-range forces from damaging the integrity of the tailored biological surface. Plasma polymer overcoating to the entire surface was applied to maintain biological compatibility while creating chemically homogenous surface chemistry. The idea of the study is relatively interesting. The title was carefully chosen, the topic of the article being suitable for Nanomaterials journal.
The Abstract is well written.
In the Introduction part the framework and hypotheses were clear. Emphasizing the scope and the novelty of the work in comparison with other studies is needed.
The Methods and Materials part is relatively good, in some places too many experimental details are given.
The Results and Discussion sections were well developed, properly combining the results and the discussions in the main text. The figures and tables were inspired and help the reader to understand better the phenomena. Still in Figure 2 there is no part E, although the text refers to it. There is only Figure 2 A,B,C,D. You also said that this technique of attaching NPs to the substrate can be used on other NP-featured surfaces, enhancing overall stability, depending on the end application requirements. Explain in details this kind of application and what do you consider to be a biological surface.
In Supporting Information it seems to me that Table S1 is similar to Table S3, and Table S2 is similar to all Table S4, Table S5 and Table S6. Is about the same NT-MDT NSG03 cantilever and the same magnitude of 10-15 , giving the same data. Please explain the differences that are not obvious.
Overall, the manuscript was very difficult to read and understand, although I have been a specialist in the field for many years. I did not understand why both types of roughness are presented in the tables, both Sa and Sq. Both will have the same stage of evolution. Other 3D texture parameters can be calculated from the AFM images, which would have been more important for the intended applications. I strongly suggest you do this, calculating for example: Sbi, Sci, Svi, Sdr...etc.
The Conclusions are well presented.
